# Biochemical and Structural Characterization of Ferulated Arabinoxylans Extracted from Nixtamalized and Non-Nixtamalized Maize Bran

**DOI:** 10.3390/foods11213374

**Published:** 2022-10-26

**Authors:** Muzzamal Hussain, Farhan Saeed, Bushra Niaz, Ali Imran, Tabussam Tufail

**Affiliations:** 1Department of Food Science, Government College University, Faisalabad 38000, Punjab, Pakistan; 2University Institute of Diet and Nutritional Sciences, The University of Lahore, Lahore 54590, Pakistan

**Keywords:** arabinoxylans, ferulic acid, ferulated arabinoxylans, HPLC, FTIR, maize bran, microstructure, monosaccharides, XRD

## Abstract

Maize bran is a good source of non-starch polysaccharides, having various bioactive compounds. In the current research, we extracted the ferulated arabinoxylans from nixtamalized maize bran (NMB) and non-nixtamalized maize bran (NNMB) and explored their biochemical composition (monosaccharides, phenolic compounds) and structural characteristics (FTIR, SEM and XRD) as well as antioxidant activity. Results showed that contents of ferulated arabinoxylans were 8.1 ± 0.04% and 6.8 ± 0.02 in NMB and NNMB, respectively. Moreover, the purity of arabinoxylans was 60.1 ± 0.8% and 57.04 ± 0.7% in NMB and NNMB ferulated arabinoxylans. Furthermore, ferulated arabinoxylans have higher arabinose, xylose and ferulic acid contents. FTIR spectra of NMB and NNMB ferulated arabinoxylans depicted the presence of polysaccharide compounds and the corresponding band was observed at 993 cm^−1^, which is due to glycosidic bond vibration. In addition, absorbance regions of arabinoxylans between 900 cm^−1^ to 1200 cm^−1^ were observed. Moreover, SEM micrographs of ferulated arabinoxylans had visible rough and porous surface morphology. Further, ferulated arabinoxylans of NMB and NNMB did not exhibit any sharp peaks in XRD graphs, attributed to their amorphous nature. However, XRD 2θ showed peaks at 20.0°, which predominantly indicated that the material has an amorphous state with small crystalline regions in the sample, which shows the presence of xylans (small and narrow sharp peaks).

## 1. Introduction

Non-starch polysaccharides are important because their moieties have the potential to improve nutraceutical and functional properties and to provide health benefits against various diseases [1,2]. Arabinoxylans are major non-starch polysaccharides of cell walls in cereal by-products [3]. Like other cereal by-products, maize bran is a milling by-product of maize and shows many functional and health-endorsing properties [4]. The key functional components of maize bran non-starch polysaccharides are arabinoxylans and phenolic compounds (especially ferulic acid). Maize bran is a natural source of dietary ferulated arabinoxylans [5]. Ferulated arabinoxylans (FAXs) affect the functional properties of cereal-based foods and also offer nutritional benefits of soluble and insoluble dietary fibers [6]. The molecular structure of FAXs show the presence of phenolic moieties responsible for their nutraceutical perspective. FAXs also have many positive health implications, including lowering glycemic index and cholesterol levels, preventing colon cancer and heart diseases and improving absorption of calcium and magnesium [2,7,8]. Arabinoxylans may promote the growth of bacteria that could be beneficial to health. FAXs may be an attractive bioactive moiety for pharmaceutical and biomedical applications. Different methods have been used to extract ferulated arabinoxylans from maize bran. Before extraction, nixtamalization is an alkaline treatment which is widely practiced in the cereal processing industries to enhance the nutritive value, chemical composition and functional properties of the cereal kernels and improve the quality of cereal products/by-products.

The current research was designed to extract FAXs from non-nixtamalized maize bran (NNMB) and nixtamalized maize bran (NMB). The biochemical composition and structural characteristics of extracted FAXS were assessed using gas chromatography–mass spectrometry (GC-MS), high-performance liquid chromatography (HPLC), Fourier transform infrared spectroscopy (FTIR), scanning electron microscopy (SEM) and X-ray diffraction (XRD).

## 2. Materials and Methods

### 2.1. Production of Maize Bran

Maize kernels (SAR-SUBZ-2019) were procured from the agricultural research institute in Quetta, Pakistan. Chemicals were procured from Scientific Store Faisalabad-Pakistan. 

In order to produce tortillas, maize must go through a process called nixtamalization, in which the grains are cooked in lime at a temperature higher than 70 °C for 5 h to 1 h 20 min, followed by a steeping or soaking period of 1 to 24 h, and then the bran is completely removed from kernels and considered as a by-product. Although nixtamalization partially solubilizes bran, it still retains a significant proportion of hemicelluloses. On the other hand, non-nixtamalized maize bran (NNMB) is produced from maize kernels through the dry milling process. After these processes, nixtamalized maize bran (NMB) and non-nixtamalized maize bran (NNMB) were milled and commercially micronized (particle size <120 µm).

### 2.2. Extraction of FAXs from Non-Nixtamalized Maize Bran (NNMB) and Nixtamalized Maize Bran (NMB)

An amount of 100 g NNMB and NMB samples were dried and defatted using a hot air oven and Soxhlet apparatus, respectively. Firstly, each sample was boiled for 1 h to gelatinize the starch, denature proteins and inactivate enzymes. According to Herrera-Balandrano et al. [8], we discarded the supernatant and then dried the sample for 12 h at 40 °C. Then, both samples were separately suspended in alkaline solution (KOH 4.5%) in a 1000 mL beaker and stirred at 100 rpm for 6 h at 30 °C using a magnetic stirrer. Residues were isolated from slurry by centrifugation at 7000 rpm for 15 min at 4 °C and the supernatant was discarded. Then, 1 L distilled water was added to the residue and centrifuged again (9000 rpm for 15 min at 4 °C) to recover the supernatant. The supernatant was precipitated in 65% (*v/v*) ethanol for 4 h at 4 °C (Figure 1). Precipitates were recovered and dried by solvent exchange (80% (*v/v*) ethanol or acetone) to give FAXs. Finally, FAXs were dried using a freeze drier and converted into powder form.

### 2.3. Monosaccharide Assay

FAXs extracted from NNMB and NMB were analyzed for monosaccharide composition. Individual neutral sugars in the extracted FAXs were determined after hydrolysis (1 M H_2_SO_4_, 100 °C, 90 min) and converted to alditol acetates as followed by Pettolino et al. [9]. Gas chromatography–mass spectrometry (GC-MS) (Agilent 6890) coupled with Agilent 5973 Mass Selective Detector was used to determine arabinose, xylose, galactose and glucose. For this purpose, a 30 m long DB-5 fused silica capillary column and film thickness of 1.0 µm and a 0.25 mm inside diameter with methyl polysiloxane polymer phase were used. The detector and injection port temperatures were 300 °C and 280 °C, respectively. Nitrogen was used as the carrier gas with a flow rate of approximately 1.5 mL/min. The GC-MS was operated in the electron impact mode with an ionization energy of 70 eV. A 2.0 µL splitless sample was injected to quantitate individual sugars. Monosaccharide experiments were repeated three times for accuracy.

### 2.4. Purity of Arabinoxylans

The purity of arabinoxylans was assessed by following the method of Herrera-Balandrano et al. [8] through the correction factor for arabinose from arabinogalactan as follows:$$\mathrm{Arabinoxylans}\;=\;\mathrm{Xylose}\;\left(\%\right)+\;\mathrm{Arabinose}\;\left(\%\right)-\left(0.7\times \;\%\mathrm{galactose}\right)$$

### 2.5. HPLC

The phenolic acids, ferulic acid and *p*-coumaric acid were analyzed with PerkinElmer 200 series HPLC equipped with a UV/VIS detector. The chromatographic separation was carried out on the reverse-phase C18 column.

The HPLC analysis for the quantification of ferulic acid and *p*-coumaric acid was carried out by following the method of Stavova et al. [10] with minor changes. In short, a reverse-phase HPLC system (PerkinElmer, Series 200, Waltham, MA, USA) column (150 mm 4.6 mm, particle size 2.7 nm) at 45 °C was adjusted to the following standard operating parameters. Acetonitrile (solvent A) and formic acid 0.5% (*v*/*v*) made up the mobile phase (solvent B). The data were collected and the obtained peaks of ferulic acid and *p*-coumaric acid were compared with the standard/reference values for quantification and confirmation.

### 2.6. Total Phenolic Content (TPC)

FAX extracts were prepared by following the method of Liu et al. [11]. First, 0.5 g FAXs was added to 50 mL distilled water, which was then heated to boiling and left for 30 min before being chilled in a freezer (−4 °C), filtered, and kept for use. TPC was determined by the method of López-Contreras et al. [12] with some amendments. An amount of 200 µL of extract, 2600 µL of distilled water, and 200 µL of Folin–Ciocalteu reagent were used. After 5 min, 2000 µL of 7% Na_2_CO_3_ was added. The process was carried out in the dark for 90 min after being agitated for 30 min. Finally, the UV-Vis spectrophotometer (Model U2020) absorbance was measured at 750 nm. The results were measured in milligrams of ferulic acid per gram of FAXs (mg FAE/g). The concentration was determined using the linear regression equation of the ferulic acid calibration curve at values ranging from 0 to 200 mg/L.
$$\mathrm{TPC}\left(\mathrm{mg}\frac{\mathrm{FAE}}{g}\right)=\mathrm{Concentration}\;\mathrm{of}\;\mathrm{ferulic}\;\mathrm{acid}\;\left(\mathrm{mg}\right)\frac{\mathrm{Volume}\;\mathrm{of}\;\mathrm{extract}\;\left(\mathrm{ml}\right)}{\mathrm{Mass}\;\mathrm{of}\;\mathrm{extract}\;\left(g\right)}$$

### 2.7. DPPH (2,2-Diphenyl-1-Picrylhydrazyl) Antioxidant Activity Assay

DPPH analysis was performed by adding 1500 μL of DPPH solution to 0.5 g of FAXs. After 30 min of reaction in the dark, absorbance was measured at 515 nm through a UV-Vis spectrophotometer (Model U2020). The results were expressed as micromoles of Trolox equivalents per gram of sample (µmol TE/g).

### 2.8. Fourier Transform Infrared (FTIR) Spectroscopy

Fourier transform infrared (FTIR) spectrum of NNMB and NMB FAXs powder samples were recorded on an Agilent Technologies Fourier transform infrared spectrophotometer (FTIR) (Cary 630 FTIR spectrometer). The examined sample was scanned by infrared, where a computer-connected detector identified it as a continuous wave, reported its spectrum, and assessed its functional units. The samples were loaded on a Smart iTX Module, and the spectra were recorded in the range of 400 to 4000 cm^−1^, using 32 scans and a resolution of 4 cm^−1^. 

### 2.9. Scanning Electron Microscopy (SEM)

FAX (NMB and NNMB) powder samples were analyzed for scanning electron microscopy without coating at low voltage (16 kv). For the purpose, scanning electron microscope Model Emcrafts (tabletop SEM cube series) was used to obtain images with a field emission gun scanning electron microscope. The surface morphology was studied using scanning electron microscopy with high-intensity electron beam. The images were obtained in secondary and backscattered electron image mode.

### 2.10. X-ray Diffraction (XRD)

XRD analyses of FAX powder samples were evaluated using an X-ray diffraction spectrometer (D8 Advance, Bruker, Billerica, MA, USA) X’Pert3 MRD XL with Cu-Kα radiation 1.5406 Å. The XRD powder patterns and samples were scanned in the 2θ range 2–60° at a step of 0.025 and a counting time of 50 s per point. 

### 2.11. Statistical Analysis

All trials were carried out in triplicates. The data obtained for each parameter were analyzed by mean squares and standard deviations and the lettering of each parameter was conducted by least significant difference (LSD) [13].

## 3. Results

### 3.1. FAX Content of NMB and NNMB

FAX contents in maize bran samples were 6.8% and 8.1% in NNMB and NMB, respectively. 

### 3.2. Monosaccharide Composition

The neutral sugar composition of FAXs from the maize bran sample (NNMB and NMB) is depicted in Table 1. The monosaccharide (xylose, arabinose, galactose and glucose) profile of FAXs showed that the xylose content in both sources is higher than other neutral sugars. Current results depicted that xylose and arabinose contents of NMB FAXs were 37.2 ± 0.7% and 25.5 ± 0.4% as compared to NNMB 34.3 ± 0.4% and 24.0 ± 0.4%, respectively. Galactose contents of FAX samples NMB and NNMB were 3.7 ± 0.1% and 2.9 ± 0.2%, respectively. The glucose content of NMB FAXs was 2.1 ± 0.1% while NNMB FAXs showed 1.8 ± 0.1% glucose content.

### 3.3. Purity of Arabinoxylans 

The purity of arabinoxylans extracted from MB and NMB was 57.04 ± 0.7% and 60.1 ± 0.8%, respectively. 

### 3.4. Antioxidant Activity and HPLC Quantification

The results regarding antioxidant activity and HPLC quantification were shown in Table 2. Total phenolic contents of NNMB FAXs and NMB FAXs were 8.8 ± 0.03 mgFAE/g and 11.2 ± 0.04 mg FAE/g, respectively. A DPPH test was conducted to determine the antioxidant activity of FAXs. The current results regarding DPPH values of NNMB FAXs and NMB FAXs were 28.5 ± 0.06 mg TE/g and 32.6 ± 0.05 mg TE/g. 

The ferulic acid contents determined through high-performance liquid chromatography (HPLC) in NMB and NNMB ferulated arabinoxylans were 3.4 ± 0.05 and 3.2 ± 0.02 mg/g, respectively. Furthermore, *p*-coumaric acid contents were 0.3 ± 0.01 mg/g and 0.4 ± 0.01 mg/g in NMB and NNMB FAXs. 

### 3.5. FTIR

The FTIR spectra of extracted FAX powder of NMB and NNMB are shown in Figure 2a and 2b, respectively. Both spectra revealed a similar configuration in their chemical structure, and the nixtamalization method did not change their molecular identity. However, the corresponding band was observed at 993 cm^−1^, which is due to C-OH banding and C-O-C glycosidic bond vibration in NMB. In addition, small shoulder peaks at 844 cm^−1^ and 1077 cm^−1^ are related to the glycosidic link and β (1→4) linkages between the sugar monomers. On the other hand, the principal band was observed at 1015 cm^−1^, which is assigned to the C-OH bend vibration in NNMB. The typical spectrum is the backbone of polymers and includes xylose-type polysaccharides. Arabinose side chain O-3 on xylose showed the band at 990 cm^−1^ to 1020 cm^−1^. The major change in intensity of the shoulders at 1149 cm^−1^ is related to the contributions of arabinose substitutions at the C3 position of the xylose concentration. Furthermore, the intensity of 990 cm^−1^ to 1150 cm^−1^ peaks decreases when A/X increases.

The bands 1650 cm^−1^ and 1533 cm^−1^ are associated with the amide I and amide II bands, respectively, and the phenolic acids have specific absorption bands in the 1500–1800 cm^−1^ range. In current spectra of NMB and NNMB FAXs, phenolic compounds including ferulic acid divided in an initial band vibration of 1521 cm^−1^ showed a strong aromatic ring and foremost absorption bands at 1692, 1620 and 1600 cm^−1^. Both spectrum peaks at 1680 and 1550 cm^−1^ NNMB and NMB FAXs are related to amide I and II, respectively. A clear observation of the FTIR spectrum of the cross-linked materials reveals that NNMB and NMB polymers are not significantly affected by the cross-linking process of arabinoxylans and ferulic acid. The current results spectrum reported that the NMB FAXs spectra band at 3281 cm^−1^ resembles the stretching of the OH groups and the band at 2922 cm^−1^ the CH2 groups, while ferulated arabinoxylans of NNMB showed the band at 2922 cm^−1^ of CH2 groups, the same as NMB spectra, but showed the stretching band of OH groups at 3272 cm^−1^.

### 3.6. SEM

The scanning electron photomicrographs of FAXs extracted from NNMB and NMB are shown in Figure 3. In the current SEM results, the micrograph presented an aggregation and scattering of microparticles in Figure 3a,b. The scattering in nixtamalized and non-nixtamalized maize bran arabinoxylans bran is due to the nixtamalization of maize bran, which loosens the strength of compact bonding of maize bran and its moieties. Moreover, it was also observed that nixtamalized arabinoxylans presented a more visible rough and irregular surface morphology with a porous surface than non-nixtamalized maize bran FAXs. However, the microparticles of nixtamalized and non-nixtamalized arabinoxylans presented an uneven structure at 3 µm magnification, which showed the presence of pores on the surface of arabinoxylans. The microstructure of the NMB FAXs was similar to the microstructure observed in the NNMB FAXs. 

### 3.7. XRD

The crystallographic study of NMB FAXs and NNMB FAXs was performed and the corresponding XRD spectrum is shown in Figure 4a, 4b. Both samples of ferulated arabinoxylans did not exhibit any sharp peaks in XRD graphs, attributed to their amorphous nature. For some applications, the material structure is also crucial; for instance, amorphous cellulose can be used as an absorbent. A broad peak at 250 is due to the polymer network in ferulated arabinoxylans. The current results of XRD 2θ showed peaks at 20.0° in both FAX samples. Additionally, with regard to the hemicellulose nature, the 2θ position of FAXs was similar at 20.0°. This predominantly indicated that the material has an amorphous state with small crystalline regions in the sample, which shows the presence of xylans (small and narrow sharp peaks). According to the previous study by Luna et al. [14], the substitution pattern of arabinoxylans which is reflected by the ratio of arabinose/xylose affected the amorphous or crystalline structure of arabinoxylans. The current study suggested that NMB FAXs and NNMB FAXs have amorphous structures. 

## 4. Discussion

Arabinoxylans are non-starch polysaccharides naturally present in cereal bran. In the current research, the exclusive property of arabinoxylans was explored, and their ability to form covalent linkage by the oxidative coupling of the ferulic acid that forms ferulated arabinoxylans was reported. The findings of the current results were according to Herrera-Balandrano et al. [8], who revealed that the contents of arabinoxylans were 4.89%, 7.17% and 8.23% at different treatments of maize bran by-products. Furthermore, the yield of arabinoxylans gum from nixtamalized maize bran was previously reported by Carvajal-Millán et al. [15], and the results depicted that maize bran gum cross-linked with ferulic acid was obtained to yield 5.5% and 13.0% at different treatment conditions.

According to our results, the arabinose/xylose ratio showed that arabinoxylans have a branched structure. The current results are similar to those of Herrera-Balandrano et al. [8], who showed that xylose, arabinose and galactose contents of alkali-extracted feruloylated arabinoxylans were 33.4%, 27.7% and 2.5%, respectively. The monosaccharide contents in the current research were lower than the results of Carvajal-Millan et al. [15], who revealed that maize bran gum contains arabinose (34 ± 0.9 g/100 g), xylose (40 ± 1.6 g/100 g) and galactose (3.2 ± 0.1 g/100 g). 

Furthermore, the purity of arabinoxylans from NMB and NNMB similar to that reported by Ayala-Soto et al. [16], who estimated the purity of arabinoxylans extracted from maize fiber (55.69%), resistant pericarp (61.12%) and susceptible pericarp (62.15%). The current results were correlated with the recently published results of Herrera-Balandrano et al. [8], who reported that the purity of arabinoxylans in nixtamalized maize bran extract was 55.58%, 56.92% and 61.16% at different alkali treatments. These results are closely related to the current research; however, the current results explored that nixtamalization improves the extraction rate of bioactive moieties and also improves the purity of arabinoxylans due to an increase in the solubility of dietary fibers.

The higher total phenolic content of FAXs showed the presence of bioactive compounds. The ferulic acid content in FAXs showed the association of non-covalent interactions between adjacent arabinoxylan chains or the formation of arabinoxylan cross-links of ferulic acid. Further, feruloylated arabinoxylans were dependent on the content of ferulic acid because it is the major phenolic compound cross-linked with arabinoxylans.

The FTIR spectrum graph for powder samples of ferulated arabinoxylans showed an absorbance region between 900 cm^−1^ to 1200 cm^−1^ that characterized the presence of polysaccharide compounds according to the results of Iravani et al. [17].

In a previous study, arabinoxylans powder showed a typical absorbance region between 1200 and 900 cm^−1^, characteristic of polysaccharides and consistent with the results from Robert et al. [18] and Kacuráková et al. [19]. Another study by Iravani et al. [17] explored that the arabinoxylans hydrogel has a similar spectrum to the arabinoxylans powder with a broad absorbance peak between 1200 and 900 cm^−1^ that characterized the polysaccharides bioactive moieties.

Recent research by De Anda-Flores et al. [20] reported that phenolic acid bands including ferulic acid are divided primarily into an absorption band, which showed strong aromatic ring vibration at 1517 cm^−1^, and secondary absorption bands 1690, 1620 and 1600 cm^−1^. Moreover, the band at 3292 cm^−1^ resembles the stretching of the OH groups and the band at 2935 cm^−1^ to the CH2 groups.

These SEM morphologic characteristics of FAXs have been exposed that were previously reported for maize by-product arabinoxylans [20].

In our view, this is the first evaluation of FAXs in X-ray powder diffraction along with FTIR and SEM. According to the previous study by Luna et al. [14], the substitution pattern of arabinoxylans which is reflected by the ratio of arabinose/xylose affected the amorphous or crystalline structure of arabinoxylans. The current study suggested that NMB ferulated arabinoxylans and NNMB ferulated arabinoxylans have amorphous structures on X-ray powder diffraction. Previous research by Patel et al. [21] reported that polysaccharides having amorphous nature were found to be more stable/resistant to heat. 

In addition, ferulated arabinoxylans extracted from maize bran could represent an industrial advantage to other polymers and are commonly used in the food and pharmaceutical industries.

## 5. Conclusions

Maize bran FAXs are natural and low-cost bioactive moieties. FAXs extracted from maize bran have higher contents of arabinoxylans. The purity of arabinoxylans was 60.1 ± 0.8% and 57.04 ± 0.7% in NMB and NNMB FAXs. Monosaccharide analyses revealed that the arabinose and xylose ratio was higher and showed the higher yield of ferulated arabinoxylans. Furthermore, ferulated arabinoxylans showed higher total phenolic contents with the higher antioxidant activity of DPPH of 28.5 ± 0.06 and 32.6 ± 0.05 µmolTE/g in NNMB and NMB FAXs, respectively. Moreover, ferulic acid contents of NMB and NNMB ferulated arabinoxylans were 3.4 ± 0.05 and 3.2 ± 0.05 mg/g, respectively. This is the first evaluation of FAXs in X-ray powder diffraction along with FTIR and SEM. Furthermore, FTIR spectrum of FAX powders showed that the absorbance region between 900 cm^−1^–1200 cm^−1^ characterized the presence of polysaccharide compounds. The peaks of aromatic protons explored the presence of esterified ferulic acid attached to arabinose side chains. The SEM microstructure of FAXs presents a rough and irregular surface morphology. XRD confirmed the presence of hemicellulose nature, and the material has an amorphous state. Based on the meticulous characterization of FAXs, a range of potential applications could be explored within the food, pharmaceutical and agricultural sectors. 

## Figures and Tables

**Figure 1 foods-11-03374-f001:**
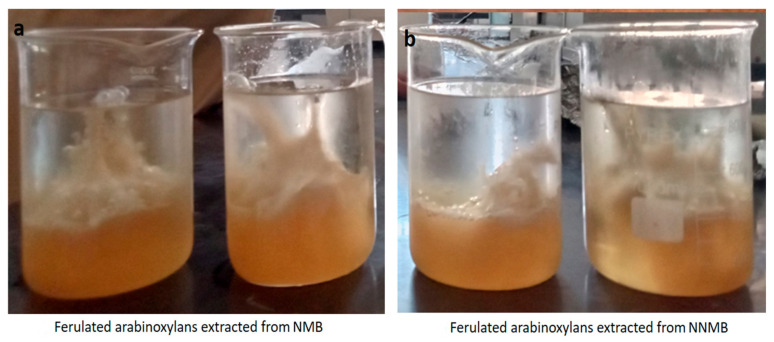
Extraction of FAXs from (**a**) NMB and (**b**) NNMB.

**Figure 2 foods-11-03374-f002:**
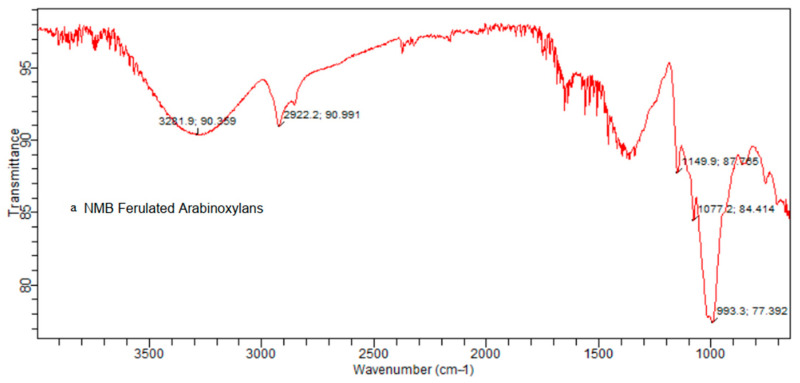
FTIR spectrum of (**a**) NMB and (**b**) NNMB ferulated arabinoxylans.

**Figure 3 foods-11-03374-f003:**
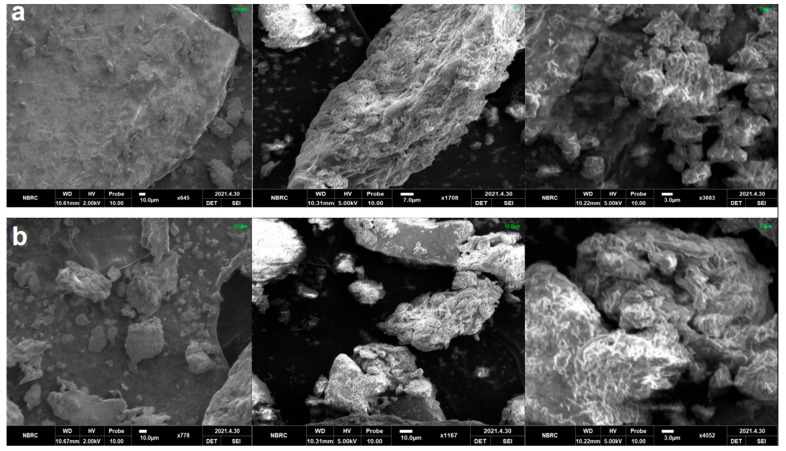
SEM microstructures: (**a**) FAXs extracted from NMB (**b**) FAXs extracted from NNMB.

**Figure 4 foods-11-03374-f004:**
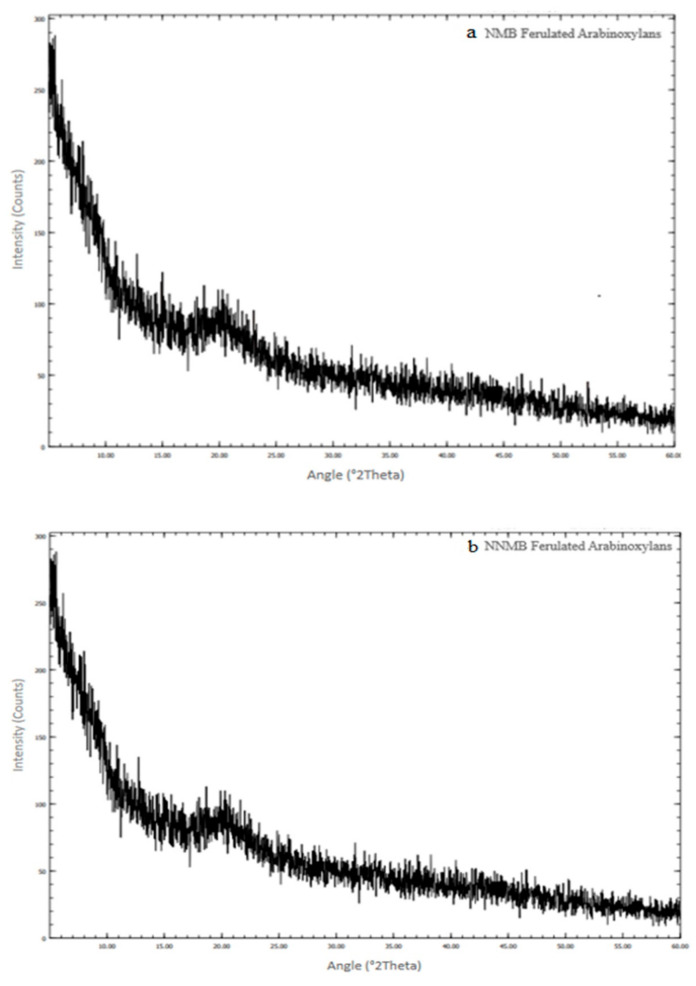
XRD spectrum of (**a**) NMB and (**b**) NNMB ferulated arabinoxylans.

**Table 1 foods-11-03374-t001:** Contents of ferulated arabinoxylans in NMB and NNMB, purity of arabinoxylans and their monosaccharide profile.

Maize By-Product Extract	Ferulated Arabinoxylans Content (%)	Arabinose (%)	Xylose (%)	Galactose (%)	Glucose (%)	Purity of Arabinoxylans (%)
NMB FAXs	8.1 ± 0.04 ^a^	25.5 ± 0.4 ^a^	37.2 ± 0.7 ^a^	3.7 ± 0.01 ^a^	2.1 ± 0.01 ^a^	60.1 ± 0.8 ^a^
NNMB FAXs	6.8 ± 0.02 ^b^	24.0 ± 0.4 ^a^	34.3 ± 0.4 ^b^	2.9 ± 0.02 ^b^	1.8 ± 0.01 ^a^	57.04 ± 0.7 ^b^

Different letters (a and b) within the column indicated the interaction of NMB and NNMB FAXs is significantly different (*p* ≤ 0.05). Results are expressed as the mean value ± standard deviation (*n* = 3).

**Table 2 foods-11-03374-t002:** Total phenolic content, phenolic acid contents and antioxidant activity (DPPH) of NMB and NNMB FAXs.

Maize Bran Extracts	Total Phenolic Content (mg FAE/g cell wall)	DPPH (µmolTE/g)	Ferulic Acid mg/g	*p*-Coumaric Acid mg/g
NNMB FAXs	8.8 ± 0.03 ^b^	28.5 ± 0.06 ^b^	3.4 ± 0.05 ^a^	0.3 ± 0.01 ^a^
NMB FAXs	11.2 ± 0.04 ^a^	32.6 ± 0.05 ^a^	3.2 ± 0.02 ^a^	0.4 ± 0.01 ^a^

Different letters (a and b) within the column indicated the interaction of NMB and NNMB FAXs are significantly different (*p* ≤ 0.05). Results are expressed as the mean value ± standard deviation (*n* = 3).

## Data Availability

Not applicable.

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
