# Peer review of "Biochemical and Structural Characterization of Ferulated Arabinoxylans Extracted from Nixtamalized and Non-Nixtamalized Maize Bran"

_foods, 2022, doi:10.3390/foods11213374_

Round 1

Reviewer 1 Report

Biochemical and Structural Characterization of Ferulated Arabinoxylans Extracted from Nixtamalized and Non-nixtamalized Maize Bran.

The main purpose of this work was extracted the ferulated arabinoxylans from nixtamalized maize bran (NMB) and non-nixtamalized maize bran (NNMB) and explored their biochemical composition and structural characteristics.

In general, this manuscript is well-written and presented relevant results. However, in my opinion, it is necessary to explain and clarify in the text why the authors studied Nixtamalized and Non-nixtamalized Maize Bran.

Page 3. Figure 1:

It is not clear in this figure 1 which is the NMR extraction and which is the NNMR. Please identify each of them in this figure.

Page 9:

Figure 4 is not visible and needs to be formatted.

Page 10:

Line 317: Delete “NMB ferulated arabinoxylans and NNMB ferulated arabinoxylans” from this sentence, because it was cited twice.

Page 11: Conclusions

Lines 339-341: In order to clarifying better the application of ferulated arabinoxylans in maize bran, the authors could have evaluated their gelling capacity. Since the Arabinoxylan gels are receiving increasing attention due to for example, characteristics such as neutral odor and taste and stability to pH. In this way, the following phrase cited in the conclusion of this article could be better explained: “Based on the meticulous characterization of ferulated arabinoxylans, a range of potential applications could be explored within the food, pharmaceutical and agricultural sectors.”

Author Response

18 October 2022

Dear Referee,  

We would like to thank the referee for the close reading and for the proper suggestions. We hope that we provide all the answers to the reviewer’s comments.

Thank you very much for the recommendations to publish our paper entitled “BIOCHEMICAL AND STRUCTURAL CHARACTERIZATION OF FERULATED ARABINOXYLANS EXTRACTED FROM NIXTAMALIZED AND NON-NIXTAMALIZED MAIZE BRAN”.

The present version of the paper has been revised according to the reviewer’s suggestions.           

We uploaded the corrected version of the article for which we used the track changes for the addition text.

GENERAL COMMENTS:Biochemical and Structural Characterization of Ferulated Arabinoxylans Extracted from Nixtamalized and Non-nixtamalized Maize Bran.

The main purpose of this work was extracted the ferulated arabinoxylans from nixtamalized maize bran (NMB) and non-nixtamalized maize bran (NNMB) and explored their biochemical composition and structural characteristics.

Referee comments: In general, this manuscript is well-written and presented relevant results. However, in my opinion, it is necessary to explain and clarify in the text why the authors studied Nixtamalized and Non-nixtamalized Maize Bran.

Response: We would like to thank to the referee for his/her close reading of our manuscript data. In current research, nixtamilization is an additionl treatment on which we have observed considerable difference in the results. Post-research results showed that nixtamilization improved the yield of ferulated arabinoxylans, improve the purity of arabinoxylans and antioxidant activity of extract. The suggestion regarding the importance of nixtamilization has been added in introduction section line 46-49.

Referee comments: Page 3. Figure 1: It is not clear in this figure 1 which is the NMR extraction and which is the NNMR. Please identify each of them in this figure.

Response: Thanks for your valuable comment. Suggestion has now been incorporated and updated figure clearly showed the NNMB and NMB extraction.

Referee comments: Page 9: Figure 4 is not visible and needs to be formatted.

Response: Thanks for your valuable suggestion. Figure 4 has now been updated/improve the visibility.

Referee comments: Page 10: Line 317: Delete “NMB ferulated arabinoxylans and NNMB ferulated arabinoxylans” from this sentence, because it was cited twice.

Response: NMB ferulated arabinoxylans and NNMB ferulated arabinoxylans” has now been deleted which was mistakenly cited twice. The typo error has now been removed at same place.

Referee comments: Page 11: Conclusions

Lines 339-341: In order to clarifying better the application of ferulated arabinoxylans in maize bran, the authors could have evaluated their gelling capacity. Since the Arabinoxylan gels are receiving increasing attention due to for example, characteristics such as neutral odor and taste and stability to pH. In this way, the following phrase cited in the conclusion of this article could be better explained: “Based on the meticulous characterization of ferulated arabinoxylans, a range of potential applications could be explored within the food, pharmaceutical and agricultural sectors.”

Response: Dear reviewer, the main objective of this study was to extract ferulated arabinoxylans from non-nixtamalized maize bran (NNMB) and nixtamalized maize bran (NMB). The biochemical composition and structural characteristics of extracted ferulated arabinoxylans were assessed using Gas Chromatographic-Mass spectrophotometer (GC-MS), High-Performance Liquid Chromatography (HPLC), Fourier transform infrared spectroscopy (FTIR), Scanning electron microscope (SEM) and X-ray diffraction (XRD).

Furthermore, the gelling capacity, neutral odor, taste, stability and pH were not the part of the current studies. The raised comments will help out in conducting the next trials that will exactly cover these characteristics.

So, we added the following conclusive sentence on the base of current results “Based on the meticulous characterization of ferulated arabinoxylans, a range of potential applications could be explored within the food, pharmaceutical and agricultural sectors.”

Finally, the authors would like to thank reviewers for their appreciations and for all the suggestions because these helped us to correct our paper and to optimize it.

Sincerely,

Muzzamal Hussain et al.

Reviewer 2 Report

This study investigates the biochemical and structural characterization of non-starch polysaccharides i.e., ferulated arabinoxylans (FAXs) from nixtamalized maize bran (NMB) and non-nixtamalized maize bran (NNMB). Monosaccharides, phenolic compounds and antioxidant activity of FAXs in NMB and NNMB were measured, and FTIR, SEM and XRD were used to explore its structural characteristics. However, the manuscript was poorly organized or written.

1. The effects of nixtamalized treatment on the biochemical and structural characterization of FAXs are required to improve the quality of the manuscript. The results and discussion should be improved. In addition, as shown in Discussion, as many syudies have been conducted to investigate the corresponding parameters of FAXs, the authors should make their innovations clear.

2. The use of abbreviation in the manuscript is confusing. For example, non-nixtamalized maize bran and nixtamalized maize bran were abbreviated to NNMB and NMB, respectively, in Line 45-46. However, the authors did a repetitive work in Line 60-63. In addition, ferulated arabinoxylans was abbreviated to FAXs (in Line 36), but the authors almost always use ferulated arabinoxylans in the manuscript.

3. The presentation of the figures and tables is confusing. For example, the citation of Figure 1 was not found in 2.2., so why it is shown in there? Besides, which is A? and which is B? In addition, Figure 1 A and B, and Figure 3 a and b or 3 a & b are found in the manuscript. Also, a footnote to the table was even found before the table.

4. Line 225: “, (a & b) respectively” can be delect.

5. Line 226: it should be “The scattering”.

6. Line 270-271: “... the oxidative coupling of the ferulic acid. Ferulated arabinoxylans.” ??

7. References should be strictly formatted.

Author Response

18 October, 2022

Dear Referee,  

We would like to thank the referee for the close reading and for the proper suggestions. We hope that we provide all the answers to the reviewer’s comments.

Thank you very much for the recommendations to publish our paper entitled “BIOCHEMICAL AND STRUCTURAL CHARACTERIZATION OF FERULATED ARABINOXYLANS EXTRACTED FROM NIXTAMALIZED AND NON-NIXTAMALIZED MAIZE BRAN”.

The present version of the paper has been revised according to the reviewer’s suggestions.           

We uploaded the corrected version of the article for which we used the track changes for the addition text.

GENERAL COMMENTS:This study investigates the biochemical and structural characterization of non-starch polysaccharides i.e., ferulated arabinoxylans (FAXs) from nixtamalized maize bran (NMB) and non-nixtamalized maize bran (NNMB). Monosaccharides, phenolic compounds and antioxidant activity of FAXs in NMB and NNMB were measured, and FTIR, SEM and XRD were used to explore its structural characteristics. However, the manuscript was poorly organized or written.

Referee comments: 1. The effects of nixtamalized treatment on the biochemical and structural characterization of FAXs are required to improve the quality of the manuscript. The results and discussion should be improved. In addition, as shown in Discussion, as many syudies have been conducted to investigate the corresponding parameters of FAXs, the authors should make their innovations clear.

Response: We would like to thank to the referee for his/her close reading of our manuscript data. In current research, nixtamilization is an additionl treatment on which we have observed considerable difference in the results. Post-research results showed that nixtamilization improve the extarction of ferulated arabinoxylans, improve the purity of arabinoxylans and antioxidant activity of extract. The reasoning of difference in current results has now been added at suitable places.

Furthermore, current reasearch work is the first evaluation of comparative and structural properties of ferulated arabinoxylans in X-ray powder diffraction along with FTIR and SEM. Moreover, current study concluded that ferulated arabinoxylans extracted from maize bran and its characterization could represent an industrial advantage aspect to other polymers and are commonly used in the food and pharmaceutical industries. We are the pioneer of such comprehensive and comparative study in Pakistan.

Referee comments: 2. The use of abbreviation in the manuscript is confusing. For example, non-nixtamalized maize bran and nixtamalized maize bran were abbreviated to NNMB and NMB, respectively, in Line 45-46. However, the authors did a repetitive work in Line 60-63. In addition, ferulated arabinoxylans was abbreviated to FAXs (in Line 36), but the authors almost always use ferulated arabinoxylans in the manuscript.

Response: Thanks for your valuable suggestions. As words are too long to carry more space owing to which we used abbrevations non-nixtamalized maize bran (NNMB) and nixtamalized maize bran (NMB). Furthermore, abbreviation FAXs has now been used throughout the manuscript.  

Referee comments: 3. The presentation of the figures and tables is confusing. For example, the citation of Figure 1 was not found in 2.2., so why it is shown in there? Besides, which is A? and which is B? In addition, Figure 1 A and B, and Figure 3 a and b or 3 a & b are found in the manuscript. Also, a footnote to the table was even found before the table.

Response: The citation of Figure 1 has now been added in 2.2. which showed the  extract precipitated in 65% (v/v) ethanol. Furthermore, tables and figurese captions have now been updated. The footnote has now been added at the end of tables.

Referee comments: 4. Line 225: “, (a & b) respectively” can be delect.

Response: The suggestion has now been incorporated.

Referee comments: 5. Line 226: it should be “The scattering”.

Response: The suggestion has now been incorporated.

Referee comments: 6. Line 270-271: “... the oxidative coupling of the ferulic acid. Ferulated arabinoxylans.” ??

Response: The incomplete sentense has now been rephrased for better understanding. “In the current research, the exclusive property of arabinoxylans was explored and reported their ability to form covalent linkage by the oxidative coupling of the ferulic acid that form asferulated arabinoxylans.”

Referee comments: 7. References should be strictly formatted.

Response: The references have now been formatted according to journal instructions.

Finally, the authors would like to thank the reviewers for their appreciation and suggestions because these helped us correct our paper and optimize it.

Sincerely,

Muzzamal Hussain et al.

Reviewer 3 Report

Reviewer comments:

I have critically and carefully evaluated this article entitled as: “Biochemical and Structural Characterization of Ferulated Arabinoxylans Extracted from Nixtamalized and Non-nixtamalized Maize Bran” and I have reached the
following conclusions:
Authors, I recommend MINOR REVISION of publication of this article in Journal Foods.

This work is well written and authors used appropriate literature.

Please find my suggestions:

Abstact:

Line 13: When You write about chemical composition you must mention chemical compounds examined in your reserach articel (RA) – monosaccharides and phenolic compounds and not antioxidant activity because antioxidant activity is biological activity of examined material... Therefore, please could you rewrite sentence in the Lines 11-14 as: „In current research, we extracted the ferulated arabinoxylans from nixtamalized maize bran (NMB) and non-nixtamalized maize bran (NNMB) and explored their biochemical composition (monosaccharides, and phenolic compounds), and structural characteristics (FTIR, SEM and XRD), as well as biological activity (antioxidant activity)“?

Also, you mentioned antioxidant activity in your abstract, could you add some results from this activity?

Keywords:

Line 26: It is more transparent if you use alphabetical order of some of the used keywords. For example: Maize bran; ferulic acid; ferulated arabinoxylans; monosaccharides; FTIR; XRD

1. Introduction

Line 41-42: please correct the end of the sentence to:“and improving absorption of calcium and magnesium ions [2,7,8]

Line 48: Please correct the name of used technique to: Gas chromatography–mass spectrometry.  This is correct name, not name which use used. Therefore, Line 68 you correct to 1 h.

2. Materials and methods

Line 58: You used hours and minutes (full names) when you first time mentioned time units. Please after that you might use h and min (abbreviated units).

Lines 73 and 75: you must used space between number and appropriate unit(s), for example: line 73: 7000 space rpm, or line 75: 9000 space rpm and 15 space min. Please check and correct throughout the whole MS.

Line 86: (1 M H2SO4, IOO°C, 90 min.) – please correct to (1 M H2SO4, 100 °C, 90 min) – in the correct version you must write number one hundred 100 space and °C, and 90 min without sign point (.).

Lines 87-88: Gas chromatography–mass spectrometry you might write in the abbreviated form GC-MS, based on the inserted abbreviation in the line 48.

Line 89: 30 space m and please check lines 90, 92, 94, and 100 (you also must correct by using space between number and unit(s))

Lines 102, 105, 110: Please could you correct p-coumaric acid to p-coumaric acid with p in italic

Line 114: 0.5 space g

Line 118: After 5 min not 5 mins

Line 118 also: Na2CO3 please correct chemical formula of sodium carbonate

Line 119: 90 min

On the page 4 (lines 114, 117, 118) You used L as abbreviation for liter, but at the begin of the text you marked as l. Please check throughout the work and correct to L.

References:

Line 395: you used different font style for Burgara-Estrella, A.J.; please correct it!

Author Response

19 October 2022

Dear Referee,  

We would like to thank the referee for the close reading and for the proper suggestions. We hope that we provide all the answers to the reviewer’s comments.

Thank you very much for the recommendations to publish our paper entitled “BIOCHEMICAL AND STRUCTURAL CHARACTERIZATION OF FERULATED ARABINOXYLANS EXTRACTED FROM NIXTAMALIZED AND NON-NIXTAMALIZED MAIZE BRAN”.

The present version of the paper has been revised according to the reviewer’s suggestions.           

We uploaded the corrected version of the article for which we used the track changes for the addition text.

GENERAL COMMENTS:

Reviewer comments:

I have critically and carefully evaluated this article entitled as: “Biochemical and Structural Characterization of Ferulated Arabinoxylans Extracted from Nixtamalized and Non-nixtamalizedMaize Bran” and I have reached the
following conclusions:
Authors, I recommend MINOR REVISION of publication of this article in Journal Foods.

This work is well written and authors used appropriate literature.

Please find my suggestions:

Referee comments:

Abstact:

Line 13: When You write about chemical composition you must mention chemical compounds examined in your reserach articel (RA) – monosaccharides and phenolic compounds and not antioxidant activity because antioxidant activity is biological activity of examined material... Therefore, please could you rewrite sentence in the Lines 11-14 as: „In current research, we extracted the ferulated arabinoxylans from nixtamalized maize bran (NMB) and non-nixtamalized maize bran (NNMB) and explored their biochemical composition (monosaccharides, and phenolic compounds), and structural characteristics (FTIR, SEM and XRD), as well as biological activity (antioxidant activity)“?

Also, you mentioned antioxidant activity in your abstract, could you add some results from this activity?

Response: Thanks for your valuable suggestions. The suggestions have now been incorporated. However, according to journal instructions; abstract should be ≤200 words. So, we choose appripriate words for readers attraction.

Referee comments:

Keywords:

Line 26: It is more transparent if you use alphabetical order of some of the used keywords. For example: Maize bran; ferulic acid; ferulated arabinoxylans; monosaccharides; FTIR; XRD

Response: The suggestions have now been incorporated. The kewords have been added in alphabetical order; Arabinoxylans, ferulic acid, ferulated arabinoxylans, HPLC, FTIR, maize bran, microstructure, monosaccharides, XRD

Referee comments:

  1. Introduction

Line 41-42: please correct the end of the sentence to:“and improving absorption of calcium and magnesium ions [2,7,8]“

Response: The suggestions have now been incorporated. The mentioned sentense has been rephrased according to your comment.

Referee comments: Line 48: Please correct the name of used technique to: Gas chromatography–mass spectrometry.  This is correct name, not name which use used. Therefore, Line 68 you correct to 1 h.

Response: The suggestions have now been incorporated.

Referee comments:

  1. Materials and methods

Line 58: You used hours and minutes (full names) when you first time mentioned time units. Please after that you might use h and min (abbreviated units).

Response: The suggestions have now been incorporated in throughout the manuscript.

Referee comments: Lines 73 and 75: you must used space between number and appropriate unit(s), for example: line 73: 7000 space rpm, or line 75: 9000 space rpm and 15 space min. Please check and correct throughout the whole MS.

Response: The suggestions have now been incorporated in throughout the manuscript.

Referee comments: Line 86: (1 M H2SO4, IOO°C, 90 min.) – please correct to (1 M H2SO4, 100 °C, 90 min) – in the correct version you must write number one hundred 100 space and °C, and 90 min without sign point (.).

Response: The suggestions have now been incorporated.

Referee comments: Lines 87-88: Gas chromatography–mass spectrometry you might write in the abbreviated form GC-MS, based on the inserted abbreviation in the line 48.

Response: The suggestions have now been incorporated.

Referee comments: Line 89: 30 space m and please check lines 90, 92, 94, and 100 (you also must correct by using space between number and unit(s))

Response: The suggestions have now been incorporated in throughout the manuscript.

Referee comments: Lines 102, 105, 110: Please could you correct p-coumaric acid to p-coumaric acid with p in italic

Response: The suggestion has now been incorporated.

Referee comments: Line 114: 0.5 space g

Response: The change has now been done.

Referee comments: Line 118: After 5 min not 5 mins

Response: The change has now been done.

Referee comments: Line 118 also: Na2CO3 please correct chemical formula of sodium carbonate

Response: The change has now been done.

Referee comments: Line 119: 90 min

Response: The change has now been done.

Referee comments: On the page 4 (lines 114, 117, 118) You used L as abbreviation for liter, but at the begin of the text you marked as l. Please check throughout the work and correct to L.

Response: The change has now been done.

Referee comments: References: Line 395: you used different font style for Burgara-Estrella, A.J.; please correct it!

Response: The style has been changed as per formatting.

Finally, the authors would like to thank reviewers for their appreciations and for all the suggestions because these helped us to correct our paper and to optimize it.

Sincerely,

Muzzamal Hussain et al.

Round 2

Reviewer 2 Report

The quality is improved.